# The Impact of Political Instability on Sustainable Rangeland Management: A Study of Borana Rangeland, Southern Ethiopia

Yeneayehu Fenetahun [1,2,3], Yuan You [1,2], Xinwen Xu [1,2], Vincent Nzabarinda [1,2,3] and Yongdong Wang [1,2,*]

1   State Key Laboratory of Desert and Oasis Ecology, Xinjiang Institute of Ecology and Geography, Chinese Academy of Sciences, Urumqi 830011, China; Tihunie.Fentahun@bdu.edu.et (Y.F.); youyuan@ms.xjb.ac.cn (Y.Y.); sms@ms.xjb.ac.cn (X.X.); vincentnzabarinda@mails.ucas.edu.cn (V.N.)
2   National Engineering Technology Research Center for Desert-Oasis Ecological Construction, Xinjiang Institute of Ecology and Geography, Chinese Academy of Sciences, Urumqi 830011, China
3   University of Chinese Academy of Sciences, Beijing 100049, China
*   Correspondence: wangyd@ms.xjb.ac.cn; Tel.: +86-139-9915-0554

**Abstract:** Political instability (PI) occurs between governments and other political elites either at the local, regional, and/or national levels. Planning, implementing, and monitoring of sustainable rangeland management strategies have a significant impact on the political environment of an area. In this study, the term PI implies an unsafe and unstable exercise of political power, and is a major obstacle to the implementation of sustainable rangeland management. The main purpose of this research was to provide empirical and theoretical knowledge by testing hypotheses about the impact of PI on the implementation of sustainable practice of rangeland management. Using in-depth interviews, this study conducted both structured and unstructured group discussions with 300 representative households of local pastoralists and others who were considered the key stakeholders in the sustainable activities of rangeland management. Results indicated that the local communities are significantly susceptible to the economic, environmental, and socio-cultural effects of sustainable management of rangeland due to PI. Furthermore, the impact of PI on the economic, environmental, and socio-cultural aspects of rangelands indicators was evaluated. The findings also proved that the satisfaction of pastoralists with rangeland productivity and function was significantly affected, and prevented pastoralists from participating in rangeland management practice.

**Keywords:** political instability; sustainability; rangeland; impact; Borana; Ethiopia





## 1. Introduction

Political instability occurs at local, regional, and/or national levels between the government and other political elites [1,2]. These actions may result in changes in the pastoralist community's subsistence patterns, disruption of traditional territorial governance arrangements, and reduced adaptive capacity of the sustainable practice of rangeland management [3]. Rangelands are natural grasslands used by domestic livestock or wild animals for grazing or browsing [4]. However, in many parts of the world, including Africa (mainly Ethiopia), the current rapid degradation of rangelands is caused by both natural and human-induced factors [5]. In Ethiopia, about 20%, 24%, and 51% of the rangelands are in good, degraded, and highly degraded status, respectively, indicating a decrease in the productivity and carrying capacity potential of rangeland [5–7]. In the rangeland area, pastoral livelihoods are mainly characterized by socio-ecological stress, risk, and uncertainties due to changing socio-political, economic, and natural environment conditions [8,9].

Sustainable rangeland management has been traditionally linked to the concepts of economic, natural, and socio-cultural environments, which include promoting economic

growth, protecting and improving pastoralists' quality of life, and increasing future opportunities by improving rangeland productivity [10]. At present, the pastoralists in the Borana rangelands of Southern Ethiopia are seriously challenged by low livestock productivity, leading to a decline in the number of livestock, severe livestock deaths during dry periods, and an increase in the number of people vulnerable to food scarcity and reliance on food aid [11,12]. Overgrazing, drought, poor management practices, increasing population numbers, and infestation of invasive plant species are the main drivers of rangeland degradation in most of the world's rangelands, including the Borana rangeland [13–15].

*Major Causes of Rangeland Degradation in the Study Site*

Overstocking, and associated overgrazing, is the major cause of rangeland degradation. It is driven by social prestige and wealth attached to livestock, in addition to population increase in the rangelands [16]. Overgrazing is defined as continuous grazing over several years that results in deterioration of the plant community and a decline in the vigor, production, and biodiversity of rangelands [17]. Climate change is also a contributor to rangeland degradation through its effects on the ecological dynamics of these systems [18,19]. Climate change is influencing pastoral mobility trends locally and in trans-border areas, as pastoralists transcend Eastern Africa borders in search of better pastures and resources [20]. This is a result of extensive droughts in different areas, causing a progressive decline in vegetation quantity and quality, and inadequacy of water [20,21]. Invasive species have attracted significant concern with respect to rangeland degradation. These species may encroach rangelands by rapidly spreading and establishing new sites [20]. The common indigenous plant species known for bush encroachment comprise the Acacia family and include *Acacia melifera, Acacia senegal, Acacia seyal, Acacia drepnolobium,* and *Commiphora africana* [22]. Bush encroachment, in combination with invasive plant species, not only suppresses forage availability for livestock but also incurs increased management costs. Agriculture and the associated developmental practices in the rangelands have increasingly contributed to rangeland degradation [23]. Agricultural activities are often associated with rangeland degradation, and newcomers to these rangelands often pursue agricultural production at the expense of pasture productivity [20].

However, the breakdown of traditional governance systems and political instability (PI) in Ethiopia has (mainly during the past three years, or from 2018 to date) become a development constraint and is having a significant impact on the rangeland resources of the environment, specifically the productivity and sustainability of the Borana rangeland ecosystem [10,24,25]. A large proportion of rangelands in Teltele are communal and these are managed using traditional governance structures that constitute and enforce norms and values of their sustainable use [20,26]. However, the implementation of these traditional institutions has decreased, weakening their capacity to manage rangeland-associated problems because of the influence of the formal government structures and the impact of the frequent occurrence of PI on the study site [26]. Many scholars argue that several policies and by-laws have significantly infringed upon the customary land rights and undermined pastoral land tenure systems that championed sustainable natural resource management [20,26]. These policies include state-sponsored resettlement schemes targeting rangelands, which are mostly perceived as vast and idle lands [20]. Changes in land tenure and resource management policies result in degradation, especially on communal rangelands [27]. The breakdown of social structures subject rangelands to the "tragedy of the commons". The tragedy of the commons describes a situation in which collective actions of some users of shared resources contravene the general good of the other users through overexploitation [28]. The free-access nature of the rangelands often makes them vulnerable to misuse, depletion, or spoiling by certain users through improper, unequal, or unfair utilization, and unsustainable agricultural practices, overstocking, and overgrazing [29]. Thus, free access to rangelands is often associated with reduced abilities to effectively control grazing and can be the cause of resource-based conflicts between pastoralists. Many studies have indicated that there have been a number of large-scale

clashes among Ethiopian pastoralists in recent decades, leading to severe human loss, causalities, family displacement, and rapid rangeland degradation [30–32]. The most-publicized of these incidents occurred in Borana rangeland (Figure 1).

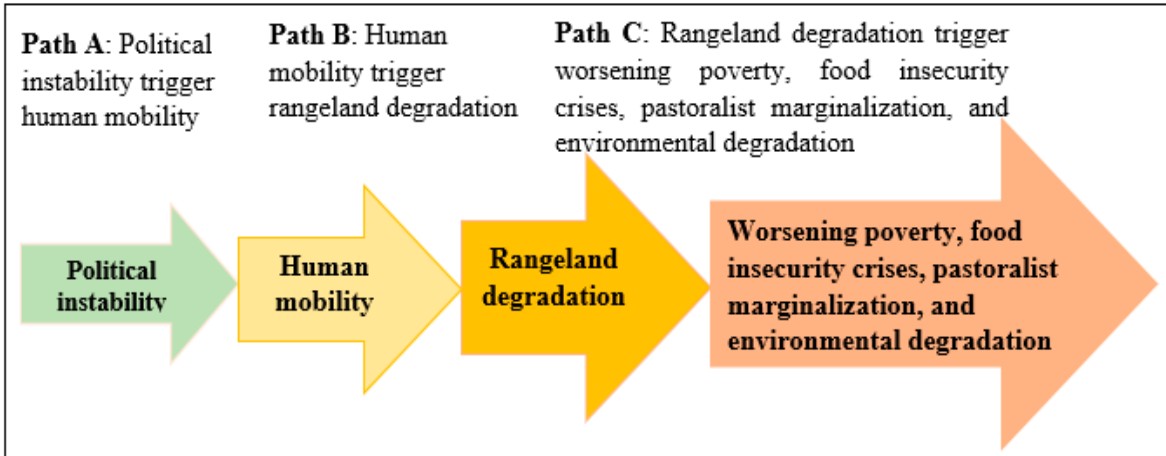

**Figure 1.** Conceptual path of the impact of political instability on pastoralist livelihood.

The government has regularly tried to monitor these pastoral areas via inter-governmental authorities to avoid conflicts within pastoral communities [33]. The "enabling political environment for rangeland management" indicator shows that rangeland sites used for grazing are more likely used to develop, implement, and follow-up management plans in areas with more stable political conditions. Conversely, lack of planning, implementation practice, and follow-up are more likely in areas with unstable political environments [34]. The term PI overlaps with governance [35]; however, in this case, PI does not refer to political parties or systems, rather it indicates the situation and practice within pastoral communities, in which leadership, structure, mechanisms, and strategies or policies are critical to the exercise and implementation of sustainable rangeland management [36,37]. In the study area, political instability is mainly due to internal conflict, youth unemployment, land ownership, military involvement in politics, religious tensions, ethnic tensions, lack of democratic accountability, corruption, inequalities, ethnic political parties, misrepresentation of historical events, and the policy of ethnic-based federalism, which has escalated old inter-ethnic resource conflicts [38,39]. This has complicated access to grazing sites in the Borana rangelands, decreased the usual practice of rangeland management, and increased the vulnerability of Borana pastoralists to both anthropogenic and natural factors that accelerate rangeland degradation. Pastoralists' perceptions of the impacts of PI on sustainable practices and productivity of rangeland management vary [40]. Thus, we assessed economic, environmental, and socio-cultural aspects using the structural model shown in Figure 2. Therefore, in our proposed model, we incorporated observations to test our hypotheses regarding the impact of PI, and pastoralists' perceptions of its effect, on rangeland management practice.

There are numerous studies available on the socio-economic impacts of PI on the Borana rangeland [41]. However, in previous studies, quantitative evaluations indicating the impact of an unstable political environment on sustainable rangeland management have not been recognized. Therefore, the impact of PI on sustainable rangeland management has not yet been assessed in research conducted at the study site. The objective of this study was to provide empirical and theoretical knowledge, via tested hypotheses, about the impact of political instability on the implementation of sustainable practices of rangeland management. To address the above-mentioned knowledge gap, we posed the following questions: (1) How does PI affect the practices of rangeland management, sustainable utilization, and value chains among pastoralists and for the country overall? (2) What are the characteristics of the instability in the rangeland, and what are the likely

future scenarios? (3) What actions are being taken by different actors to reduce the impact of PI? Or how are these actors adapting? The following three hypotheses were simultaneously developed and tested in the current study: (1) Political instability has a significant effect on sustainable rangeland management; (2) political instability has a direct effect on the economic, environmental, and socio-cultural aspects of the pastoral communities; and (3) pastoralist dissatisfaction due to instability has an impact on the participation of rangeland management practice.

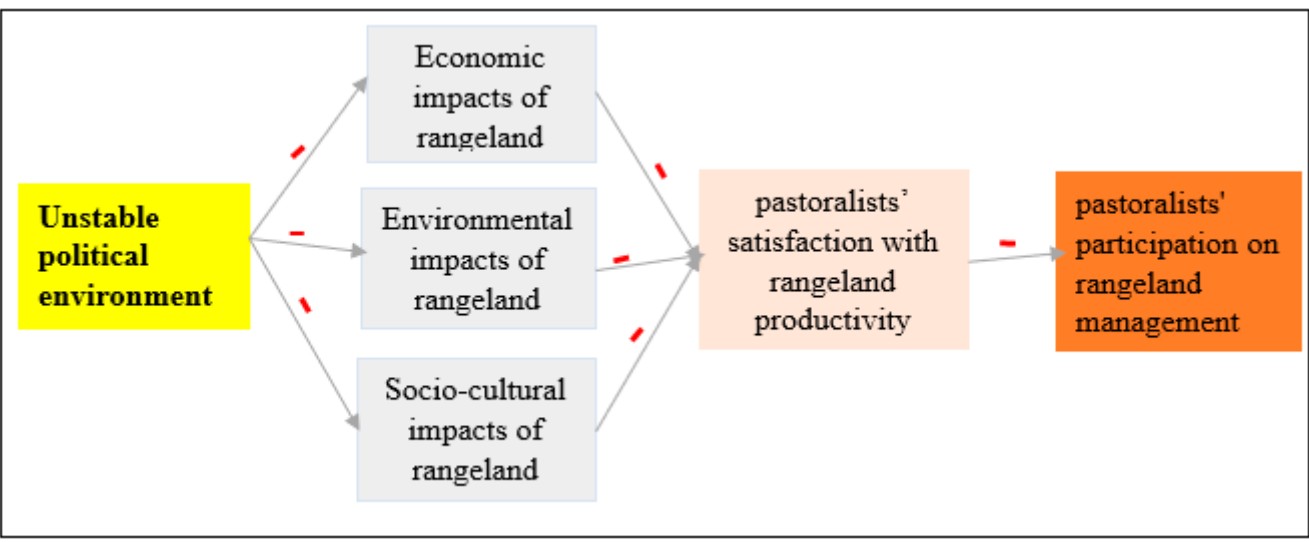

**Figure 2.** Theoretical model of impacts and perceptions of political instability.

## 2. Materials and Methods

### 2.1. Study Area

The study was conducted in the Yabelo district of the Borana zone, Southern Ethiopia (Figure 3), covering an area of 1,543,000 ha, 68% (1,049,240 ha) of which is rangeland [25]. The site was selected because it is one of the major rangeland sites in which frequent political instability has been observed and, therefore, the rangeland management practice and pastoral communities in this area are the most vulnerable to PI. The area is located on the Addis to Moyale road, 566 km south of Addis Ababa (capital city of Ethiopia), and lies between 4°30′55.81″ and 5°24′36.39″ north latitude and between 7°44′14.70″ and 38°36′05.35″ east longitude [25,42]. The altitude is about 1000 to 1500 m, with a maximum elevation of 2000 m. Rainfall is bi-modal, with the main rainy season (73%) occurring between March and May, and the short rainy season (27%) occurs between September and November [42]. The mean annual rainfall recorded ranges from 450 to 700 mm [43] and the mean annual temperature varies from 19 to 24 °C, with little seasonal variation. The potential evapotranspiration varies from 700 to 3000 mm [44]. The main soil types in the study area include red sandy loam soil, black clay, volcanic light-colored silt clay, and silt [42]. Based on the relative coverage, the dominant grass species in the investigated sites include *Chrysopogon aucheri*, *Chloris roxburghiana*, *Cenchrus ciliaris*, *Harpachne schimperi*, and *Cyperus bulbosus* (greater than 10% both in the dry and wet seasons) [45].

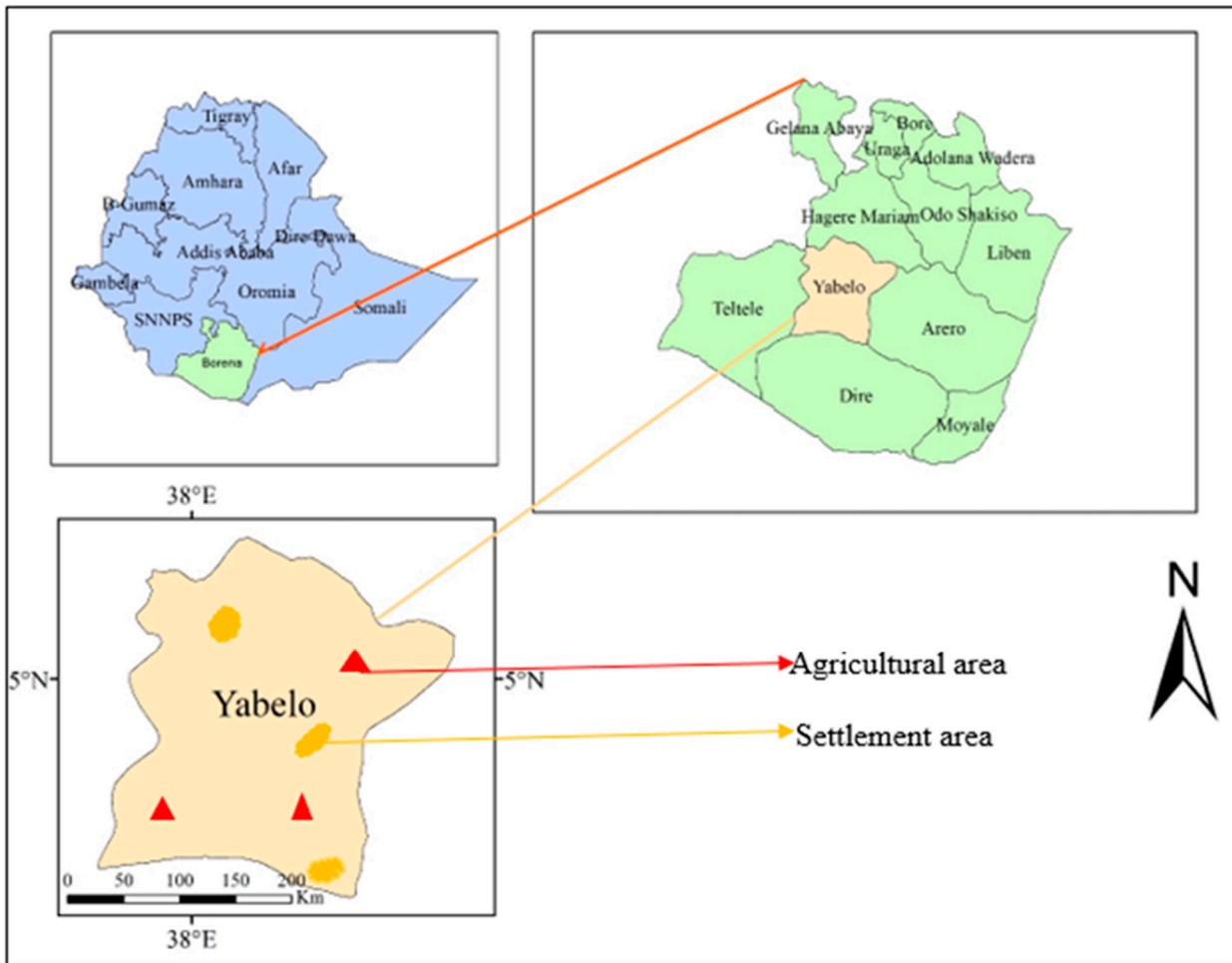

**Figure 3.** Location of the study area.

According to the latest national census reported in 2017, this district has a total population of 100,501, including 56,246 males and 44,255 females; 6289 (14.2%) of its population are urban dwellers. The main economic activities are livestock breeding (80%), and agriculture. According to data reported by the zone livestock office, the estimated total number of all livestock species is 244,134, and the numbers and proportions (of the total livestock population grazed in the study area) of each species found in the study district are: cattle (104,000, 42.6%), goats (86,039, 35.2%), sheep (37,591, 15.4%), camels (13,305, 5.4%), horses (138, 0.06%), mules (159, 0.07%), and donkeys (2902, 1.2%). Furthermore, political instability has been the major cause of pastoral community displacement and degradation of Yabelo rangelands, especially during the last three–four years.

*2.2. Data Sources and Methods*

This study combined in-depth field visit surveys (to obtain comprehensive information from the target population about the current research topic and their interest), question-naires, and rangeland use policy analysis linked to the source of change in rangeland vegetation. Data collection was conducted over a period of two months, from November to December 2020. Based on the literature, we first listed the sustainable rangeland management indicators in order to develop our assessment parameters. All of the listed indicators were included in the questionnaires and used in the subsequent survey. The self-administered questionnaire was distributed to Yabelo pastoralist communities. A total of 300 individuals (200 males and 100 females), including 30 stakeholders (20 males and 10 females) from different government sectors (from agricultural, livestock, justice, environment, and biodiversity institutes), who had lived in the study district for 10–20 years,

were selected. The total sampling population depended primarily on our time and budget, however, we aimed to select a sample that was representative of the study area. To obtain more accurate opinions and perceptions, the prepared questionnaire for each occupied household was translated to the local language (Amharic), and appropriate words were provided for the terms "political instability" ("Yepolitcal almergagat" in Amharic) and "impact" ("yemiyasektlew gudat"). We then interviewed and discussed the impact of PI on sustainable rangeland management practices and associated vegetation change based on interviewees' observations and experiences in the district. The interview questionnaire consisted of both structured and unstructured questions and focused essentially on the sustainable rangeland management indicators.

To develop our measurement tools, a list of sustainable rangeland management indicators was evaluated by relevant field experts from higher officials to the district stakeholder level. These experts agreed on the major 17 indicators used to evaluate the dynamic change of the rangeland. The questionnaires were designed to acquire basic information such as gender, age, ethnic group, level of education, household income, current employment sector, number of people engaged in livestock farming in the family, and livestock farming income rate. Information such as the perceptions of respondents regarding the impact of the unstable political environment on the Borana rangeland from economic, environmental, and sociocultural aspects, pastoralists' satisfaction with rangeland functions, and their participation in rangeland sustainable management activities were also recorded (Tables 1 and 2).

**Table 1.** Distribution of pastoralists demographic characteristics in Borana rangeland (*n* = 300).

| Characteristics | Frequency | Percentage |
|---|---|---|
| Gender | | |
| Male | 200 | 66.7 |
| Female | 100 | 33.3 |
| Age (years) | | |
| Young (20–35) | 111 | 37.0 |
| Middle (36–52) | 147 | 49.0 |
| Elder (≥52) | 42 | 14.0 |
| Ethnicity | | |
| Amhara | 14 | 4.7 |
| Oromo | 262 | 87.3 |
| Tigray | 5 | 1.7 |
| Others | 19 | 6.3 |
| Education | | |
| Primary school (Grade 1–8) | 152 | 50.7 |
| High school (Grade 9–12) | 66 | 22.0 |
| Tertiary education (college and university) | 33 | 11.0 |
| Not educated | 49 | 16.3 |
| Occupation | | |
| Livestock rearing | 249 | 83.0 |
| Employment | 43 | 14.3 |
| Business | 93 | 31.0 |
| Labor work | 17 | 5.7 |
| Annual income ($) | | |
| Below 500 | 67 | 22.3 |
| 500–1000 | 188 | 62.7 |
| 1000–1500 | 28 | 9.3 |
| Above 1500 | 17 | 5.7 |

**Table 1.** *Cont.*

| Characteristics | Frequency | Percentage |
|---|---|---|
| Annual household income rate from rangeland (%) | | |
| 0 | - | - |
| 1–25 | 27 | 9.0 |
| 26–50 | 39 | 13.0 |
| 51–75 | 91 | 30.3 |
| 76–100 | 143 | 47.7 |

Notes: 1$ = 39 Ethiopian birr (ETB).

**Table 2.** Results of pastoralists' data about indicators of sustainable rangeland management (*n* = 300).

| List of Indicators | Responses in % | | | | |
|---|---|---|---|---|---|
| | SA | A | N | DA | SDA |
| Political Instability | | | | | |
| PI-1 Rangeland management activities is less concerned by the local and national government. | 78.3 | 17.0 | 0.7 | 2.3 | 1.7 |
| PI-2 The local rangeland areas has frequently expose for interest conflict. | 22.3 | 66.0 | 4.3 | 3.0 | 4.4 |
| PI-3 Local pastoralists are claimed about fair utilization of rangeland. | 27.0 | 56.3 | 4.0 | 7.3 | 5.4 |
| PI-4 Key rangeland areas monopolized by a few politically powerful people and their families. | 18.0 | 77.7 | 1.0 | 2.0 | 1.3 |
| Impacts on economic aspect | | | | | |
| IE-1 The gap between the rich and poor in the rangeland area increased. | 38.7 | 49.0 | 9.3 | 3.0 | - |
| IE-2 Living costs of the community doubled. | 44.7 | 55.3 | - | - | - |
| IE-3 Mobilization of livestock's restricted and caused hunger and less productivity. | 22.3 | 73.0 | 1.0 | 3.0 | 0.7 |
| IE-4 Freely movement is become under risk and make the life more difficult those who have hand-to-mouth way of life. | 53.7 | 45.0 | 1.3 | - | - |
| Impacts on environment | | | | | |
| EI-1 Frequent political instability impact on the life both plant and animals (through high displacement of pastoralists, pollution etc.). | 12.7 | 83.7 | 1.0 | 2.3 | 0.3 |
| Impacts on socio-cultural aspect | | | | | |
| SCI-1 Political instability can cause loss for local culture and social interaction. | 36.7 | 59.0 | 2.0 | 0.7 | 1.6 |
| SCI-2 Political instability limits cultural exchanges and infrastructure access. | 17.7 | 82.0 | 0.3 | - | - |
| Pastoralists satisfaction with rangeland productivity | | | | | |
| PS-1 I am not satisfied with the rangeland management activities. | 26.0 | 53.0 | 5.7 | 13.7 | 1.6 |
| PS-2 I am not satisfied with pastoralists involvement and follow-up in the planning and implementation of management practice in the area. | 20.7 | 67.3 | 4.3 | 7.0 | 0.7 |
| PS-3 I am not satisfied with the rangeland productivity and equitable benefits from it. | 38.0 | 60.7 | 1.0 | 0.3 | - |
| Pastoralists participation on rangeland management activities | | | | | |
| PP-1 I do not participate in any rangeland management activities. | 48.0 | 23.0 | 10.3 | 3.7 | 13.0 |
| PP-2 I do not participate in any awareness creation program about how to conserve rangeland in a sustainable way. | 26.7 | 44.3 | 1.3 | 22.0 | 5.7 |
| PP-3 I do not participate in the fair and equal sharing of benefits from rangeland. | 23.7 | 56.0 | 4.3 | 11.7 | 4.3 |

Notes: SA = strongly agree, A = agree, N = neutral, DA = disagree, SDA = strongly disagree.

*2.3. Data Analysis*

The data were analyzed based on the indicator variables with principal component analysis (PCA). Based on the impacts and perceptions of PI, the model was developed and then incorporated into the confirmatory factor analysis (CFA). CFA is a method for measuring latent variables. It extracts the latent construct from other variables and shares the greatest variance with related variables [2,5]. Finally, structural equation modeling (SEM) was used to establish the connections between these factors [2]. SEM is a powerful, multivariate technique found increasingly in scientific investigations to test and evaluate multivariate causal relationships [2]. SEMs differ from other modeling approaches because they test the direct and indirect effects on pre-assumed causal relationships. SEM follows logical steps such as model specification, identification, parameter estimation, model evaluation, and model modification [2,37]. The model specification defines the hypothesized relationships among the variables in an SEM based on one's knowledge. In our study, SEM confirmed the connections between the impact of political environment, economic, environmental, and sociocultural impacts of rangelands pastoral dissatisfaction, and nonparticipation in sustainable rangeland management. Cronbach's alpha and Kaiser-Meyer-Olkin (KMO) statistics tests for consistency, reliability, and validity between measurement variables were performed prior to factor analysis [46]. Using IBM SPSS Amos 25.0 software Armonk, NY, USA, the confirmatory factor analysis (CFA) was carried out and the hypothesized relationships between the constructs were tested using $p$ and $t$ values (Table 3).

**Table 3.** The path coefficients between paired constructs (associated with impacts and perceptions of political instability) in Borana rangeland.

| Paired Constructs | | | *t*-Values | *p*-Values |
|---|---|---|---|---|
| Economic impacts | <— | Unstable political environment | −19.621 | ** |
| Environmental impacts | <— | Unstable political environment | −3.477 | * |
| Socio-cultural impacts | <— | Unstable political environment | −19.839 | ** |
| Pastoralist's satisfaction | <— | Economic impacts | −6.670 | ** |
| Pastoralist's satisfaction | <— | Environmental impacts | −2.702 | * |
| Pastoralist's satisfaction | <— | Socio-cultural impacts | −2.747 | * |
| Pastoralist's participation | <— | Pastoralist's satisfaction | −8.209 | ** |

Note: ** = statistically significant at $p < 0.001$, * = Statistically significant at $p < 0.05$.

## 3. Results

*3.1. Socio-Demographic Characteristics of the Respondents*

Table 1 shows that the majority of respondents were male (66.7%), and most of them were from the Oromo ethnic group (87.3%), while only 4.7%, 1.7%, and 6.3%, were Amhara, Tigray, and other ethnicities, respectively. The ages of respondents were 36–52 years old (49.0%) and 20–35 years old (37.0%), followed by the lowest proportion of elder respondents aged above or equal to 52 years old (14.0%). The proportion of respondents who attended school or college (considered primary, high, and tertiary education) was 83.7% and the largest number of respondents (50.7%) were those who attended primary education (grade 1–8) followed by high school completed (22%); 16.3% of respondents did not attend any education level (uneducated). The major sources of income or occupations of the local

communities were livestock rearing (83%), employment (either governmental or non-governmental sectors) (14.3%), business (31.0), and labor work (5.7%). This indicates that livestock rearing is the main source of income for the community in the rangeland areas. Table 1 also shows that more than half of the respondents (62.7%) have an annual household income of $500–1000, followed by a family annual income of less than $500, accounting for 22.3%. The numbers of respondents with annual household incomes above $1000 and $1500 were the lowest, accounting for 9.3% and 5.7%, respectively. With regard to the rangeland household income rate, the respondents whose rangeland income accounts for 0% of the annual income were zero, indicating that all of the communities living in the Borana rangeland area were directly or indirectly benefiting from the rangeland. Other rangeland income ranges of 1–25%, 26–50%, 51–75%, and 76–100% were 9.0%, 13.0%, 30.3%, and 47.7%, respectively. In general, we can conclude from the above analysis that although political instability is one of the most significant bottleneck problems in the Borana rangeland, respondents of the Borana pastoral community were directly or indirectly involved in the rangeland, and all have a rangeland revenue source (Table 1).

### 3.2. Analysis of Sustainable Rangeland Management Indicators

The descriptive statistics and measurement model results of rangeland sustainable management indicators based on the respondent's data are described in Table 2 and Figure 4.

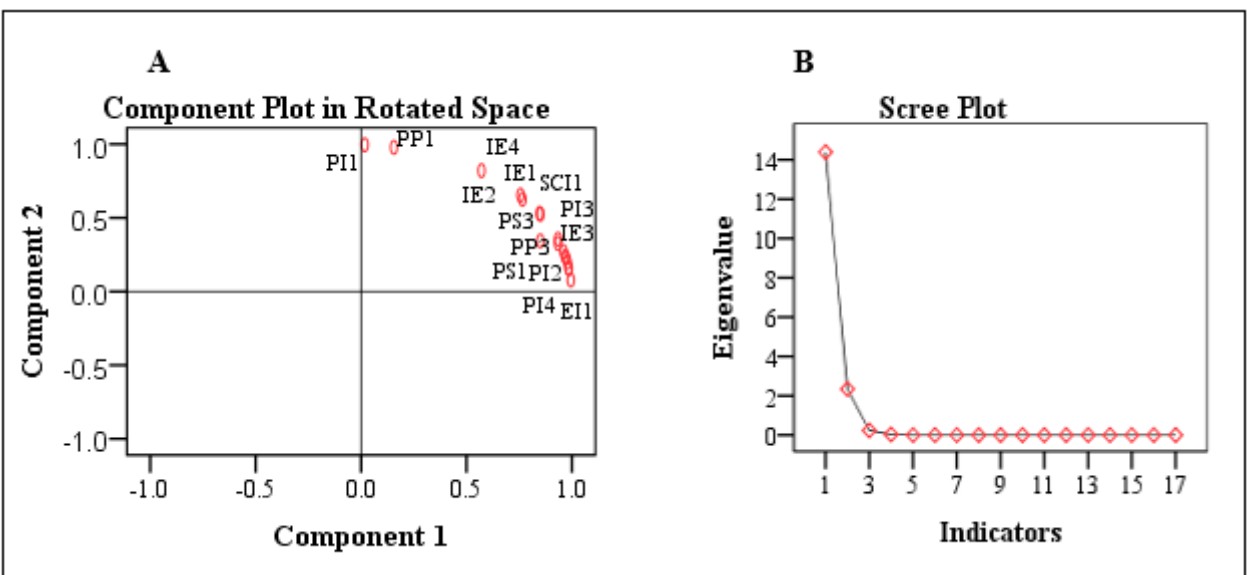

**Figure 4.** Indicator's linkage in principal components in a two-dimensional specie (**A**) and scree plot; eigenvalues plotted in descending order (**B**).

The reliability of the formulated questionnaires based on the assessment indicators was evaluated using Cronbach's alpha test. Using the prepared questionnaires, this test indicated the consistency of the information obtained through repeated measurement regarding the same issue [47,48]. The reliability analysis was conducted using SPSS and all assessment indicators addressed in the questionnaires were scaled. If the value of the reliability scale coefficient is between 0.8 and 0.9, it is considered to have very good reliability, and if between 0.7 and 0.8, it is considered to be reliable [2]. The Cronbach's alpha coefficients of all assessment indicators in this study were in the interval of 0.7–0.9, with the majority having a value greater than 0.82, indicating that the reliability of the majority of indicators was very good, and that the scales and reliability tests could be accepted (Figure 4). The effectiveness of the assessment indicators used in the evaluation was validated on the basis of both content (by logic analysis) and structural (by factor analysis) aspects. Validity refers to the degree of accuracy of the assessment indicators in addressing the items of interest [49]. For all of the assessment indicators, the KMO values of all items

were greater than 0.7 at a significant level ($p < 0.05$), indicating that all assessment indicators addressed using the questionnaires were more effective (Figure 5). Using principal component analysis (PCA), the relationship of indicators related to sustainable rangeland management was evaluated. The correlation matrix of each indicator related to the impact of PI in the economic, environmental, and socio-cultural aspects was analyzed and explained (Figure 4A). The plotted eigenvalues were obtained from the correlation matrix and variation was also calculated and explained by the components (Figure 4B). Component loadings with varimax rotation, in addition to the eigenvalues, showed that all components had eigenvalues greater than one (Figure 4B). All listed indicators occurred at component one, which indicates that there was a positive correlation between all of the listed indicators regarding sustainable rangeland management.

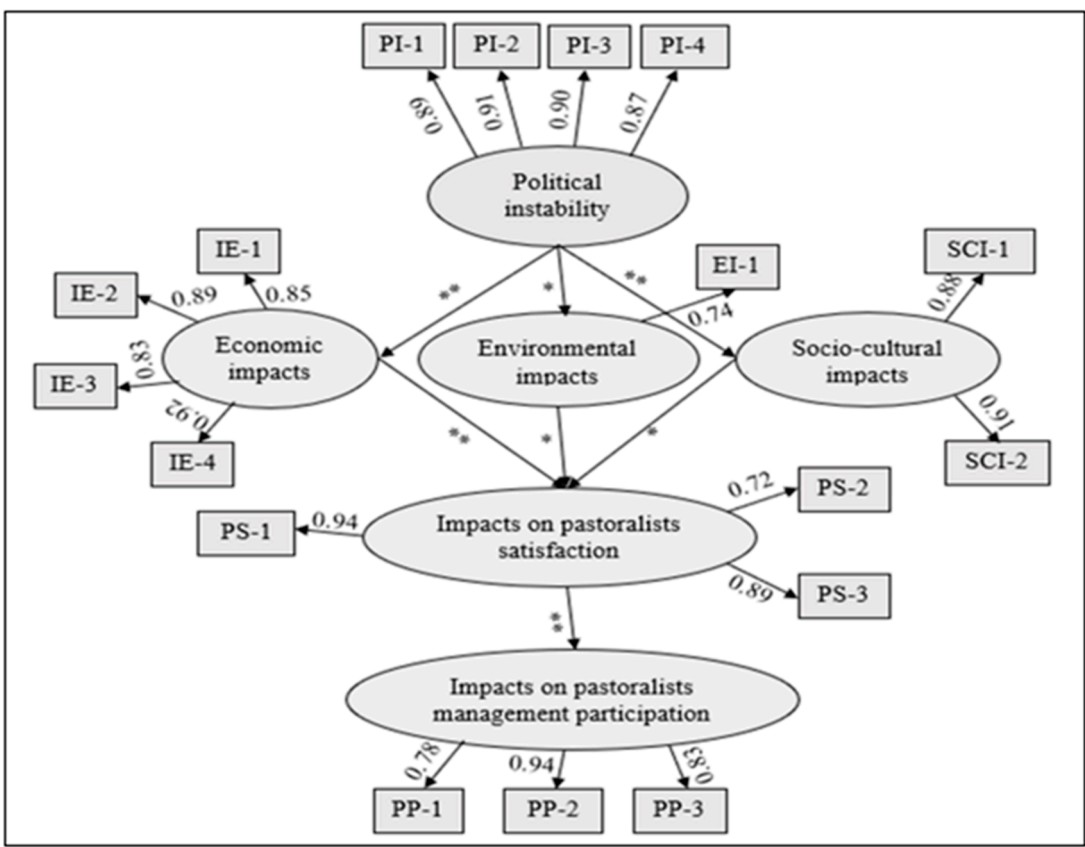

**Figure 5.** The model of political instability effects on sustainable rangeland management. ** Statistically significant at $p < 0.001$, * statistically significant at $p < 0.05$ (the model was modified from [30]).

### 3.3. Impact Analysis of Political Instability

Using structural equation modeling (SEM), the hypothesized impact of PI on the sustainable rangeland management approach was tested. The model was used to provide information on the linkage between the indicators and the associated impacts and perceptions of political instability. The model shows the connection between unstable political environment, economic impacts, environmental impacts, socio-cultural impacts, pastoralist satisfaction with rangeland productivity, and pastoral participation in rangeland management activities (Table 3). A significant value within the paired constructs was shown by most of the path coefficients.

The structural part of the model (Table 3) showed that PI has a significant negative effect on pastoralists' economic ($t = -19.621$, $p < 0.001$), environmental ($t = -3.477$, $p < 0.05$) and socio-cultural ($t = -19.839$, $p < 0.001$) aspects in the rangeland area, and this negatively affects the pastoralist's satisfaction ($t = -6.670$, $p < 0.001$; $t = -2.702$, $p < 0.05$; $t = -2.747$,

$p < 0.05$) due to economic, environmental and socio-cultural impacts, respectively. This indicates that the pastoralist participation ($t = -8.209$, $p < 0.001$) in rangeland management activities has a significant negative effect. Thus, the occurrence of PI was shown to have a multidimensional impact on the sustainable management practice and functioning of rangeland. Therefore, all of our hypotheses (H1, H2, and H3) were proven and found to be acceptable.

The six major proposed impacts associated with PI in the proposed model were shown to have a true and significant effect on sustainable rangeland management and its functions based on the evaluation indicators (Figure 4).

## 4. Discussion

The model developed in the current study focused on six interrelated factors, namely PI (as a major driving factor), three direct effects due to rangeland PI (economic, environmental, and socio-cultural effects), pastoralists' satisfaction with rangeland functions, and pastoralists' participation in sustainable rangeland management (Figure 4) [29,30]. The impact of PI on the economic, environmental, and socio-cultural aspects of sustainable rangeland management can be clearly understood from the model: (1) the direct effects of the three aspects of sustainable rangeland management on the satisfaction with rangeland productivity and overall function of local pastoral communities; and (2) the indirect effects of PI on local pastoral satisfaction with rangeland, and direct effects of those pastoralists' satisfaction with rangelands on pastoralists' sustainable management activities for rangeland [16,20,50–52].

Our study confirmed that the three hypotheses related directly to the impact of PI on the conservation practice of rangeland. A total of 17 sustainable rangeland management indicators identified six major factors, representing a self-standing construct model linked to the description shown in Tables 2 and 3. This data contributes to the dynamics of rangeland and local pastoralists' awareness by integrating the impacts of the current PI into the sustainable rangeland management approach, and can be used to survey the involvement of local pastoralists in rangeland management activities [23,26,27]. Therefore, the proposed assessed parameters in our model regarding the rangeland status (Figure 4) and overall discussion (Table 2) begin with the impact of PI, which was assessed by four indicators: local and national government are less concerned with rangeland management activities, local rangeland areas have frequently exposed conflicts of interest, local pastoralists have misinformation about fair use of rangeland, and key rangeland areas are monopolized by a few politically powerful people and their families [17]. Then, using our proposed model, we analyzed both the direct and indirect impacts of the six major factors [30].

Based on the data obtained from the respondents, we concluded that the Borana rangelands pastoral community is highly aware of the impact of PI, because most respondents agreed with statements used to evaluate PI and its indicators' effects. According to the respondents' data, the majority agreed (strongly agree (SA) + agree (A)) on the existence of the rangeland assessment indicators listed: rangeland management activities are less concerned both at the local and the national government level (95%), rangeland is frequently exposed to a conflict of interest (88.3%), pastoralists are alleged to have fair use of rangeland (83.3%), and key rangeland areas are monopolized by a few politically powerful individuals and their families (95.7%), clearly proving the direct impact of PI on economic, environmental and socio-cultural aspects [26,30]. The effects of PI had a major impact on economic and socio-cultural pastoralist interactions, revealing the high level of internal consistency of the construct ($-19.621$ and $-19.839$). In addition, three direct connections were hypothesized with respect to the impact of PI on sustainable rangeland management in the Borana rangeland, based on the different perspectives. The first hypothesis (H1) assumes that PI has a significant effect on sustainable rangeland management. The result confirms that the majority of respondents do not participate in any rangeland management activities (71%). Because of the current PI situation, respondents do not participate in any awareness creation program for the management and rehabilitation of

degraded rangelands (71%) or in the fair and equal sharing of rangeland benefits (79.7%) (Table 2) [2,30].

The second hypothesis (H2) assumes that PI has a direct effect on the pastoral communities' economic, environmental, and socio-cultural aspects. From an economic perspective, our outcome proved that almost all respondents agreed that: the gap between rich and poor people in the rangeland area has increased (87.7%), the community's living costs have doubled (100%), and mobilization of livestock has restricted and caused hunger. Less productivity (95.3%), and free movement is at risk and makes life more difficult for those with a hand-to-mouth way of life (98.7%) (Table 2). From an environmental perspective, our results showed that almost all respondents (96.4%) confirmed that both plants and animals have a frequent PI impact on life, including their functions (through high displacement of pastoralists and animals, and pollution). The impact of PI on the socio-cultural aspects of pastoralists was statistically significant, indicating that nearly all respondents agreed that PI can cause culture and social interaction losses (95.7%), and limit cultural exchanges and access to infrastructure (99.7%). The third hypothesis (H3) assumes that there is an impact on rangeland management practice involvement resulting from pastoralists' dissatisfaction due to PI. The results confirmed that pastoralists' dissatisfaction with rangeland functioning due to current PI had a significant impact on the active and volunteer participation of the communities in sustainable rangeland management activities (Table 3). Our study generally affirmed that the impact of PI could be linked either directly or indirectly to the dynamics of rangeland functions in socio-cultural, economic, and environmental aspects [2,5,30,53].

Figure 4 shows that all of the indicators evaluated in this study were highly reliable and were significantly affected by the current political situation that was the subject of this study. It is therefore assumed that the perception that PI for sustainable rangeland management influences economic development, environmental protection, and rich socio-cultural resources in the Borana rangeland results in dissatisfaction with rangeland and livestock rearing activities in their communities [20,21]. This high rate of dissatisfaction results in low participation in the pastoral community management of rangeland. Based on this investigation, a conclusion can be drawn that although the direct reason for the weak participation of pastoralists in the rangeland management practice was due to the dissatisfaction of the local community with the rangeland productivity, the unstable political environment in the Borana rangeland area was one of the primary indirect causes for this weak participation [2–4,41]. If local communities, authorities, and government officials desire to listen to the voices of pastoralists, and to provide them with a safe and stable political environment in their daily lives, their participation will a major requirement. Local pastoral communities will then be able to engage in programs of conservation and sustainable management based on what they believe and want, and within the scope of the local government's awareness and resources [33]. To summarize, the political structure, power centralization, and tendency of political practice in the rural areas of many underdeveloped African countries such as Ethiopia, especially in pastoralist and semi-pastoralist areas, is a detrimental issue for local communities [20,54]. Therefore, unstable political environments (hiding preferential policies, unequal participation opportunities, and unequal sharing of benefits), will reduce their interest in participating in rangeland management activities [16,18]. One of the main prerequisites is the active participation of local pastoralists in the conservation, sustainable management, and utilization of rangelands in their settlement regions. In this case, the main requirement that needs to be implemented and practiced is a safe and stable political environment. The local pastoralists play a critical role in implementing effective sustainable rangeland management measures because they are familiar with indigenous practical knowledge, including of the requirements to achieve protection and when it should be implemented.

## 5. Conclusions

Previously, no data had been reported about quantitative evaluations of PI, indicating the impacts of an unstable political environment on sustainable rangeland management have not been recognized. In particular, the impact of PI on sustainable rangeland management had not yet been assessed at the study site. This study aimed to assess the impact of PI on the sustainable management of rangelands. The involvement of pastoralists in rangeland rehabilitation and conservation strategies is strongly influenced by the political environment in the communities in the rangeland areas. The evaluation results confirm that the current PI at the study site has had a significant effect on the pastoralists' livelihoods, in terms of all economic, environmental, and socio-cultural aspects. These impacts may result in the pastoral communities' dissatisfaction with the productivity and general function of the rangeland. As a result, pastoralists' participation in sustainable management activities in the rangeland is highly influenced by the poor political environment management in the rangeland area. Accordingly, the assessment result of the current study helps both the local pastoral communities and the government to understand and reassess the impact of PI on the rangeland and livelihoods of the pastoral communities, and the nation in general.

Based on the measured indicators and impacts, the active participation of local communities in sustainable rangeland management activities can be increased by creating a smooth and peaceful dimension in the political environment. Therefore, in order to improve sustainable rangeland management and its productivity in the Borana rangeland, the following measurement activities are suggested. First, the local communities, and governmental and non-governmental organizations, should attempt to raise awareness and promote peaceful political exercise in the region. Second, the rangeland area should be demonstrated to be more productive and to benefit from the implementation of sustainable conservation strategies. Third, comprehensive awareness of the negative impact of political instability on the economic, environmental, and socio-cultural aspects of pastoralists' life should be demonstrated, and equal opportunities created for participation in management activities and to benefit from sharing the rangeland without discrimination according to race, ethnicity, religion, or political view. To achieve the above recommendations, the political environment should incorporate researchers from different backgrounds who are involved in improving the community lifestyle and satisfaction, because local communities are one of the key stakeholders in the formulation of a stable political environment. This requires the current political system to be restructured in order to effectively practice and implement rangeland management policies based on coordination and cooperation between all stakeholders.

In the case of the arid and semi-arid zone of the Borana rangeland, it is essential to critically evaluate, follow-up, and take corrective measures to reduce the impact of the political system and power structure on the sustainability of the livestock industry. Therefore, it is important to have a clear understanding of political issues, the interests of key political actors, and approaches to mitigate personal interests, to promote and maintain sustainable livelihoods and rangeland development in this developing country [16]. The survey conducted in this study did not investigate the perceptions of different stakeholders (e.g., government and non-government authorities, local and foreign tourists, investors, private sectors involved in rangeland activities). As a result, from a different perspective, a broad view of PI impact may not have been assessed. In this regard, future research may require the inclusion of respondents from different sectors and the testing of the indicators referred above, particularly the poorly-developed impact indicators, on the political environment.

**Author Contributions:** Y.F. collected available data and contributed to writing up, gap assessment, and design. Y.Y., Y.W., V.N. and X.X. performed editing and proofing, provided important advice, and supervised the whole work during this project. All authors contributed to the article. All authors have read and agreed to the published version of the manuscript.

**Funding:** This study received financial support from the African Great Green Wall Adaptation Technical Cooperation Research and Demonstration (2018YFE0106000), Science and Technology Partnership Program, Ministry of Science and Technology of China (grant No. KY 201702010), Integration and application of appropriate technologies for desertification control in Africa (grant No. SAJC202108) and International Cooperation and Exchanges NSFC (grant No. 41861144020), as well as the support from the CAS Key Technology Talent Program.

**Institutional Review Board Statement:** Not applicable.

**Informed Consent Statement:** Not applicable.

**Data Availability Statement:** All the data generated or analyzed during this study are included in this published article and publicly available without restriction.

**Acknowledgments:** The authors wish to thank the Xinjiang Institute of Ecology and Geography, University of Chinese Academy of Sciences, and CAS-TWAS fellowship program for providing funding and the Ph.D. Scholarship for the first author. The authors greatly thank the local community and stakeholder of the Yabelo district for providing us with help and information throughout our research.

**Conflicts of Interest:** The authors declare no conflict of interest.

## Abbreviations

| | |
|---|---|
| PI | Political instability |
| PCA | Principal component analysis |
| CFA | Confirmatory factor analysis |
| KMO | Kaiser–Meyer–Olkin |
| SEM | Structural equation modelling |

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
