# Peer review of "The Impact of Political Instability on Sustainable Rangeland Management: A Study of Borana Rangeland, Southern Ethiopia"

_agriculture, doi:10.3390/agriculture11040352_

Round 1

Reviewer 1 Report

In the manuscript the authors presented an important point, although the manuscript has some drawbacks.

Main remarks:

1. Literature review - after the "introduction", there should be a part of "literature review".

2. Conclusions -  in your conclusions, please also answer the following questions:
• what are the directions for the future?
• what are the research gaps?
• what is new to this manuscript?

Author Response

We are very grateful for the reviews provided by the reviewer of this manuscript. We heartfully thank you for all your positive comments and suggestions. The comments are encouraging and the reviewers appear to share our judgment that this study and its results are ecologically important and we tried to address all the comments point-by-point on the attached file. 

Reviewer 2 Report

This is a very interesting paper dealing with the impact of political instability on sustainable rangeland management, taking as a case study the Borana rangeland in the southern part of Ethiopia. Before it is publishable the article needs some reworks, so I would like to suggest the authors to revise the following issues:

1) Ethiopia should be also a keyword.

2) The aims of the paper must be clearly presented in the introduction. Also, the introduction has to present what this paper brings new to the existing theories on political instability and sustainable rangeland management

3) The literature review could be enlarged, because at the moment the paper has only 42 cited references. This shows to the reader a pretty scarce documentation on the current theories related to political instability and sustainable rangeland management. For instance, authors can mention how remote sensing is an important dimension in pastoralism/rangeland management studies (see doi: 10.14358/PERS.69.6.675) and can look on broader pastoralist issues and find how for instance shepherds can defend their rights and rangelands (doi: 10.1080/1070289X.2017.1400322), while for political instability connected to rangelands and animal rights in different countries authors can read papers in Land Use Policy journal and in other pastoralist-connected and area-specific journals and mention for instance how rangelands could be sustainable managed in different other contexts, how different cattle or other animals (even dogs including shepherd dogs) were sometimes culled in periods of political instability (see doi: 10.1111/area.12155).

4) The method and data interpretation are nicely presented, but the discussions should better connect to the existing literature review. So how are the results in this study could be bridged to the existing literature on political instability and sustainable rangeland management? In the discussions I found only three places speaking of five connections to the current literature on the paper's topic, which are few.

Author Response

We are very grateful for the reviews provided by the editors and each of the external reviewers of this manuscript. We heartfully thank you for all your positive comments and suggestions. The comments are encouraging and the reviewers appear to share our judgment that this study and its results are ecologically important. Please see below, in blue our detailed response to each comment.  

Reviewer 3 Report

Thank you for an interesting article on a highly important topic. My main issue with the manuscript is the way you present the notion of political instability. It is useful to give a short description of how political instability is understood in the literature generally and for rangeland management more specifically, as you have done. However, the more developed description (which is better) comes far too late and is not consistent with the way you describe how you use the term in the beginning of your introduction and in the abstract (which is not adequate, and should be modified). The more developed general description should be moved up to help frame your work.

For your study, the notion of political instability has a completely different meaning however. It is operationalised through whichever way your respondents understood the term, in the interviews you conducted etc. This needs to be clearly stated, and consistently followed throughout the manuscript. Since this is the central notion in your study, you should also provide sufficient detail on the language or languages used for the study, and which exact wording in those languages was used for the term "political instability" in your interviews and questionnaires.

A second important point is to be more specific about what rangeland degradation involves concretely in the particular region of your study. While rangeland degradation is a major problem globally and across Africa, the forms it takes as well as drivers can vary locally. Please include a well-developed section describing rangeland degradation for your case.

Thirdly, you argue that political instability affects rangeland management adversely and that this in turn leads to rangeland degradation. The argument is reasonable, but you need to add some detail on the causal mechanisms and relationships between specific rangeland management practices and specific types of degradation.

I have marked some wording that needs to be reconsidered in the attached file, but the text contains a number of grammatical errors that need to be corrected before publication.

best of luck with revising and developing this manuscript!

Author Response

We are very grateful for the reviews provided by the editors and each of the external reviewers of this manuscript. The comments are encouraging and the reviewers appear to share our judgment that this study and its results are ecologically important.

And the points indicated by the reviewer and the response (blue) for each point are listed herein below.  

Round 2

Reviewer 1 Report

Accept in present form. Good luck!

Author Response

Dear reviewer,

Thank you for your time, energy, and all efforts that you did for our paper to make it scientifically sounds full and make our effort fruitful.

We all are very thank full for all your effort s and valuable comments.

Thank you.

Yeneayehu Fenetahun

On behalf of all authors. 

Reviewer 2 Report

Authors have improved their paper during the revision by adding some details in the introduction and literature review. After reading attentively the revised paper, I consider a very minor revision is needed, because I spotted two issues in my previous comments which have not been mentioned in the revised paper: 1) the importance of pastoralist identity (see O'Brien Thomas et al, 2019,  in journal Identities, volume 26, issue 4: 470-488) should be briefly mentioned, because we live in an era of modernism and climate change and the peril for pastoralists would be to loose their cultural identity; 2) animal (cattle, sheep, sheep dogs etc) rights is an important issue connected to pastoralism. As I previously mentioned, authors can briefly give some examples in the world, for instance they can mention that dog culling actions including sheep dogs in a European country (see an article in journal Area, 2015, volume 47, issue 2, pp155-165) or give other few examples of cattle who were unethically treated in several countries in the world.

Author Response

The comments raised by the reviewer are very critical and valuable for such animal-related manuscripts. And according to the objective of our paper, we tried to address the issue in the attached file.

Note: if still have any unclear point we are too happy to make it clear and get your valuable comments that help to make our paper more scientifically.

Thank you.   
